# The Optogenetic Revolution in Cerebellar Investigations

**DOI:** 10.3390/ijms21072494

**Published:** 2020-04-03

**Authors:** Francesca Prestori, Ileana Montagna, Egidio D’Angelo, Lisa Mapelli

**Affiliations:** 1Department of Brain and Behavioral Sciences, University of Pavia, 27100 Pavia, Italy; francesca.prestori@unipv.it (F.P.); ileana.montagna01@universitadipavia.it (I.M.); dangelo@unipv.it (E.D.); 2IRCCS Mondino Foundation, 27100 Pavia, Italy

**Keywords:** cerebellum, optogenetics, sensorimotor system, non-sensorimotor functions

## Abstract

The cerebellum is most renowned for its role in sensorimotor control and coordination, but a growing number of anatomical and physiological studies are demonstrating its deep involvement in cognitive and emotional functions. Recently, the development and refinement of optogenetic techniques boosted research in the cerebellar field and, impressively, revolutionized the methodological approach and endowed the investigations with entirely new capabilities. This translated into a significant improvement in the data acquired for sensorimotor tests, allowing one to correlate single-cell activity with motor behavior to the extent of determining the role of single neuronal types and single connection pathways in controlling precise aspects of movement kinematics. These levels of specificity in correlating neuronal activity to behavior could not be achieved in the past, when electrical and pharmacological stimulations were the only available experimental tools. The application of optogenetics to the investigation of the cerebellar role in higher-order and cognitive functions, which involves a high degree of connectivity with multiple brain areas, has been even more significant. It is possible that, in this field, optogenetics has changed the game, and the number of investigations using optogenetics to study the cerebellar role in non-sensorimotor functions in awake animals is growing. The main issues addressed by these studies are the cerebellar role in epilepsy (through connections to the hippocampus and the temporal lobe), schizophrenia and cognition, working memory for decision making, and social behavior. It is also worth noting that optogenetics opened a new perspective for cerebellar neurostimulation in patients (e.g., for epilepsy treatment and stroke rehabilitation), promising unprecedented specificity in the targeted pathways that could be either activated or inhibited.

## 1. Introduction

The first reports on cerebellar structure and the first hypothesis on function pointed out its high degree of connectivity with the rest of the brain and the possible relationship with higher-order functions [1]. However, since the nineteenth century, with the work of Flourens (who provided the first descriptions of the motor syndrome now called ataxia [2]), the cerebellum has been investigated for its role in motor functions, while the role of its extensive connectivity to other brain areas was overlooked. The shift in the cerebellar paradigm is only relatively recent and can be traced back to Schmahmann’s first reports of the Cerebellar Cognitive Affective Syndrome ([3] and the introduction of the Dysmetria of Thought concept in 1998 [4]. Since then, several studies have reported cerebellar abnormalities or lesions at the core of cognitive dysfunctions [5,6,7] as well as higher order function impairments associated with motor syndromes involving the cerebellum [8,9]. While the research on the cerebellar role in sensorimotor integration can rely on a well-defined anatomy and easily measurable motor outputs, investigating the cerebellar role in non-sensorimotor functions proved challenging. To find the role of the cerebellum in these cases, one needs to take into account cerebellar connectivity with other brain regions, such as the hippocampus or the neocortex, that are not usually direct. Moreover, while considering cognitive functions, the involvement of brain cortical associative areas makes the identification of the specific role of the cerebellum hard. Over the last decade, the revolution of technical approaches in neurophysiological investigations has provided new tools and granted unprecedented resolution in determining the effects of neuronal types and single pathways in complex behaviors. In particular, optogenetics proved to be a game-changing tool that neuroscience needed in order to achieve stimulation specificity, in vitro and in vivo, that was not foreseeable only twenty years ago. This technical revolution affected cerebellar research too, allowing huge steps in understanding the cerebellar contribution to both motor and non-motor functions. This review will focus on the use of optogenetics in the cerebellum to analyze behavior, to different extents. Concerning the section about the role of cerebellum in the sensorimotor system, the focus will be on the unprecedented level of detail that optogenetics allow to achieve in terms of specific neuronal types or pathways involved in precise aspects of movement. The non-sensorimotor section will summarize the main findings of the last decade, where the use of optogenetics allowed us to tackle the role of cerebellum in functions and behaviors usually associated with other brain areas. Importantly, this optogenetic revolution in cerebellar investigations might be key in achieving cerebellar stimulation (or inhibition) as a clinical treatment for a broad spectrum of disorders. 

In this review, we will provide a focused and critical summary of recent advances in cerebellar optogenetics and its applications to study neuronal circuits in physiological or pathological conditions, highlighting the advantages of photostimulation techniques, as well as emerging questions and future perspectives.

## 2. Brief Overview of the Cerebellar Anatomy and Microcircuits Organization

The cerebellum is part of the hindbrain, located above the brainstem, and is characterized by two lateral hemispheres and a medial region called the vermis. The whole cerebellum is organized in the deep cerebellar nuclei (DCN), enclosed in the white matter, and in the three layers characterizing the cerebellar cortex. Throughout the whole cortex, the network organization is repeated, with very few differences among regions. In essence, the first set of inputs is conveyed by the mossy fibers, making excitatory synaptic contacts with granule cells and Golgi cells in the granular layer (the inner cortical layer of the cerebellum). Granule cells are excitatory neurons that make synaptic contacts (through the ascending axon or the parallel fibers) with the Golgi cells, molecular layer interneurons, and Purkinje cell (PC) dendrites (in the molecular layer, the external layer of the cerebellar cortex; in between, PCs soma originate the Purkinje cell layer). Interestingly, granule cells are the only excitatory neurons in the cerebellar cortex (except for unipolar brush cells in specific regions), giving rise to an inhibitory organization in feedback and feedforward loops that endow the cortical processing with peculiar features (for a detailed review of this topic see [10,11,12]). Another set of inputs comes from the climbing fibers, originating in the inferior olive. Climbing fibers directly contact PCs, which provide the sole output of the cerebellar cortex. PCs axons make inhibitory synaptic contacts with DCN neurons. The DCN are composed of three sets of nuclei (from medial to lateral): the fastigial, the interpositus (that includes the globose and emboliform nuclei in primates), and the dentate nuclei. Both mossy and climbing fibers send collaterals to the DCN before entering the cerebellar cortex. Therefore, the DCN can compare and integrate the input signals conveyed to the cerebellum and the result of cortical processing of those same inputs. Several studies report that this anatomical organization of cerebellar microcircuits can be divided into modules, operating in loops with the regions of origin of their inputs, being the inferior olive or the rest of the brain [13]. The cerebellar role in both sensorimotor and non-sensorimotor functions likely relies on the brain regions involved in these loops. This is also evident from the recent advances in cerebellar development studies, though this topic is beyond the scope of this review (see for details on this subject [14,15]).

## 3. Pros and Cons of Optogenetics

The obvious advantage of optogenetics is the possibility to modify neuronal activity in a cell-specific or region-specific manner. This is usually achieved using promoters for proteins expressed in specific neuronal subtypes or by localizing the viral injection to confined areas. The second main advantage is the possibility to activate or inhibit the target neurons. Previously, neuronal inhibition was achieved by local lesions, pharmacological tools, or decreasing the temperature in the brain region of interest. All of these approaches have evident disadvantages. Electrical lesions are usually not precisely localized and are not reversible. Optogenetic inhibition of neuronal activity can be achieved with millisecond precision and brief duration, if needed. The pharmacological tool lacks temporal precision, and the reversibility might not be complete. The drop in local temperature is usually achieved by perfusing cold solutions, lacking both temporal and spatial precisions. The high spatial and temporal resolutions of optogenetic modulation of neuronal activity, together with the specificity of cell types and pathways, led to a definitive advance in the study of neuronal activity and connectivity on the basis of complex behaviors and pathological conditions. This was even more evident in the last few years, where technological improvements made miniaturized devices available, allowing us to modify the activity of selected neuronal types and specific connections during a wide range of behaviors and behavioral tasks, providing causal relationships between the optogenetically targeted neurons and the observed behavior.

Though the pros of optogenetics are impressive, some technical issues need to be taken into account. First of all, the need to inject viral constructs when genetically modified animals are not available. Different viral batches may come with different titer and, in every case, the dilution and the volume of the injections must be calibrated each time. This is true even when using the same batch but changing injection location, since different brain areas show different infection levels with the same construct. Moreover, the specificity of the cell type is allowed only where the target cells have unique markers, not common to other cells in the surrounding areas. This condition is not always achievable. Finally, the amount of opsins expressed on neuronal membranes might differ from trial to trial, making the net effect of light-driven neuronal activation/inhibition difficult to reproduce consistently. Nevertheless, the pros of optogenetics use in neuroscience mostly overcomes the cons, and its unique features need to be taken into account when designing experiments and interpreting results. Before proceeding, another critical issue is worth pointing out. Optogenetics is indeed a revolutionary method to excite or inhibit neurons, since it involves the opening of ion channels on their membranes, generating ion fluxes to modify membrane voltage. This condition is very similar to the physiological processes involved during neuronal activity, but it is essential to keep in mind that optogenetic stimulation is very different from the physiological condition. It is not possible to have control over the amount of currents induced in a single neuron or to affect only those neurons that are physiologically activated together by a common pathway. Every neuron expressing the opsins will react when illuminated, therefore activating or inhibiting entire regions. This condition is quantitatively different from physiological activation in terms of current amplitude in single neurons and the number of neurons affected. Though this specification was necessary, a stimulation method acting directly on neurons and mimicking the exact physiological activation is not available at the moment.

Indeed, optogenetics has moved cerebellar research ahead by allowing us to disentangle intertwined pathways at a spatiotemporal precision unmatched by other techniques. In particular, optogenetic tools allowed for (1) the genetic specificity and anatomical strategies controlling the electrical activity of selected cerebellar neurons, (2) the targeting of projections between cerebellum and other brain regions by delivering light to opsin-expressing axons, (3) the link to data obtained from in vitro/in vivo electrophysiological recordings and targeted cerebellar neurons during behavioral tasks and (4) closed-loop interventions in which optical cerebellar stimulation is guided by real-time readouts of ongoing activity [16]. 

In the following sections, the use of optogenetic tools to dissect the cerebellar role in behavior is divided for investigations of the sensorimotor and non-sensorimotor functions.

## 4. Sensorimotor Functions 

The renowned function of the cerebellum is the integration of sensorimotor information. Despite the first reports on this topic dating back almost two centuries ago, the underlying physiological mechanisms remain incompletely defined still at the neuronal and microcircuit levels. The recent development of optogenetics boosted neurophysiological research, providing a suitable tool to investigate the impact of specific neurons or pathways on behavior. Indeed, optogenetics is now primarily employed to stimulate specific components of the cerebellar circuit in order to investigate the cerebellar contribution to perception and motor control. The cerebellum is involved in loops with the cerebral cortex, including motor and sensory areas. Inputs coming to the cerebellum send collaterals to the DCN before entering the cerebellar cortex. The DCN are in the position to integrate the “raw” signal sent to the cerebellum with the results of the cortical processing of the same input. In turn, the DCN convey the integrated signal back to the brain regions of origin. These sensorimotor cortico-cerebellar loops play a crucial role in the fine control of voluntary movements [17,18].

The following sections briefly summarize the main findings made possible by optogenetics in the investigation of the cerebellar role in sensorimotor control and learning.

### 4.1. Sensorimotor Integration and Voluntary Movement

Cerebellar contribution to sensorimotor integration is particularly evident in the rodent whisker system. Cortical sensory and motor information converge at the cellular level in the cerebellar cortex (as the lateral part of Crus I), which forms a closed functional loop with the whisker motor cortex. Indeed, optogenetic stimulation of this cortical area in awake CD1-*L7*-ChR2-YFP mice resulted in alterations of ongoing movements and touch events against surrounding objects [17]. In particular, selective photostimulation of PCs of the crus I area receiving motor inputs produced inhibition of the DCN that resulted in a post-inhibitory activation of the cerebellar-thalamo-cortical pathway, yielding motor cortex activation. Lesions at the cerebro-cerebellar loop level in rodents impaired whisking-linked behavior, though not affecting the ability to move the whiskers and touch surrounding objects [17]. Interestingly, the same study reported that optogenetic perturbation of cerebellar processing led to suppression of coherent activities in the gamma band between motor and sensory cortexes. Investigating cerebellar integration of sensory and motor information might, therefore, be essential not only for the study of movement control, but also for a deeper understanding of the sensory process itself.

### 4.2. Associative Learning (Eyeblink Conditioning)

The cerebellum is known to be involved in eyeblink conditioning (EBC), which is often used in investigations involving the cerebellum in physiological and pathological conditions. In EBC, a neutral conditioned stimulus (CS, either visual or auditory) anticipates an aversive unconditioned stimulus (US), usually an air puff delivered to the eye or a peri-orbital shock, see [19,20], that causes the animal to close an eyelid out of protection. After multiple presentations of CS and US with the same temporal pattern, the animals learn to associate the CS with the following US, resulting in the closing of the eyelid when the CS (neutral stimulus) is delivered, anticipating the aversive stimulation (US). CS and US signals are conveyed to the cerebellum through distinct input pathways, the mossy fibers and the climbing fibers, respectively [21]. Both these inputs project to the cerebellar cortex and send collaterals to the DCN, where learning is thought to take place [19,21,22]. So far, investigations on EBC mechanisms have focused mainly on the role of the cerebellar cortex and, in particular, of long-term depression and long-term potentiation at the fiber parallel to the PC synapse [23,24,25,26,27,28], intrinsic plasticity of PCs [25,29], and their dependence on the external feedback provided by the climbing fibers [30,31]. Indeed, it has been shown that electrical stimulation of mossy fibers is able to substitute the sensory CS to drive eyelid conditioning in rabbits [32]. Optogenetic stimulation has the advantage of avoiding tissue damage provoked by current injection and allows for the precise activation of neuronal subpopulations or specific pathways. For these reasons, optogenetics was applied in several studies investigating cerebellar role in associative learning. A recent study extensively applied this technique in order to investigate the impact of behavioral states on associative learning, the pathways and neuronal types involved [33]. The CS was mimicked as optogenetic activation of cerebellar mossy fibers, using Thy1-ChR2/EYFP transgenic mice expressing channelrhodopsin 2 (ChR2) in cerebellar mossy fibers (MF-ChR2 mice). In particular, the authors show that locomotion is able to enhance associative learning (using delay EBC as a model), exploiting optogenetics to selectively activate mossy fibers terminals (mimicking the CS) in a specific region of the cerebellar cortex involved in eyelid movements. Their results showed that locomotor contribution to eyelid closure during the task is likely to rely on mechanisms involving the mossy fiber pathway (CS) and downstream. Optogenetics was further used to address the contribution of cerebellar neuronal types, showing that the processing of the input downstream of the mossy fibers is deeply involved in the associative learning process and that the different neuronal types impact differently on the timing of the response (with milliseconds precision). It is evident that optogenetics was key to obtain this kind of factorization of the process underlying associative learning in the cerebellum.

The cerebellar circuit underlying EBC mechanisms was further characterized, addressing the role of the internal feedback between the DCN and the cerebellar cortex within the same modules [34]. Projection neurons in the DCN send excitatory fibers that enter the granular layer to originate the so-called nucleocortical mossy fibers. The terminals of these fibers make direct synaptic contact with granule cells and Golgi cells, and provide indirect inhibition to PCs through the parallel fibers—molecular layer interneurons pathway. Gao and colleagues obtained optogenetic control of nucleocortical mossy fibers activity by injecting the AAV-hSyn-ChR2-eYFP in the interposed nuclei (Figure 1). At the same time, this procedure allowed them to label mossy fibers rosettes in the granular layer originating from nucleocortical projections. Therefore, an optical fiber implanted in the cerebellar cortex of mice used for EBC tests was sufficient to modulate nucleocortical mossy fiber activity, while the YFP labelling was used to unravel structural modifications of these terminals after learning. This protocol allowed them to characterize cortical activity during the behavioral test (using extracellular recordings mainly from PCs) and the synaptic strength of the connections to granule cells and Golgi cells (using patch-clamp recordings in vitro). These complex experimental settings allowed them to unravel a specific role of nucleocortical projections during EBC in enhancing PCs inhibition, most likely preferentially activating granule cells, in which parallel fibers activate molecular layer interneurons more than PCs (Figure 1B). In untrained animals, the optogenetic activation of nucleocortical mossy fibers did not induce the conditioned eyeblink response, while in mice where associative learning already took place, optogenetics was able to modify the strength and the latency of the response. The authors concluded that the nucleocortical pathway acts as a gain amplifier in the learned EBC response, while it cannot be sufficient to generate this form of associative learning. These observations might be generalized, suggesting the role of nucleocortical cerebellar connections to be components of an internal amplification loop that might be involved in mechanisms underlying associative motor learning [30,31,35]. Here, optogenetics proved to be indispensable to select nucleocortical fibers among the mossy fiber bundle that enters the granular layer. Electrical stimulation of the interposed nuclei could have achieved a similar effect in the cortex, but it would have activated all the downstream connections outside the cerebellum (a condition that is clearly detrimental when working with awake animals).

Recently, a new approach to EBC studies combined optogenetics and pharmacological tools in awake and freely moving rats [36]. Again, the CS was substituted with the optogenetic activation of mossy fibers (in this case, following the injection of the construct pAAV 2/9-hSyn-ChR2-mCherry in the pontine nuclei). The pharmacological approach was used to block both excitatory and inhibitory inputs to the pontine nuclei, in order to “isolate” the learning process in the cerebellum. These experiments showed that the cerebellum in itself is sufficient to generate associative learning in the form of simple EBC, though extra-cerebellar inputs are required to facilitate such learning [36].

### 4.3. Eye Movements in Monkeys

The efficacy of optogenetic stimulation in rodents opened the possibility of a potential application of this technique in a wide variety of animal models. In monkeys, PCs are known to play a crucial role in the execution of accurate movements, although their impact is still poorly understood, due to the lack of techniques for selective manipulation of their spiking activity. A recent investigation applied optogenetics to modify PC firing, using an AAV-*L7*-ChR2 construct that is specific for this neuronal type, in rhesus macaques [37]. The efficacy of the expression in this animal model was properly confirmed using immunohistochemical analysis and neurophysiological recordings. To test the impact of photoactivation of PCs on behavior, an optical fiber was implanted in the oculomotor vermis to deliver light pulses (453 nm) after saccade initiation on randomly interleaved trials (50%). Interestingly, the optical stimulation failed to evoke saccades, although it was observed to provoke saccade dysmetria. This observation confirmed, for the first time, the feasibility of genetic manipulation techniques in primates. It is worth noting that previous studies reported the ability to evoke saccades in primates using electrical stimulation to activate PCs in the oculomotor vermis [38,39,40,41], showing that the effects of electrical stimulation are stronger than optical ones, at least on primate behavior [42,43,44,45,46]. This might be explained by considering that optical stimulation exerts its effects on nearby neurons [47], which is specific for PCs, and ChR2-expressing axons generate only low frequency spiking when activated [48,49]. Nevertheless, optogenetics proved to be a valuable tool for studies in monkeys, not only in rodents. Indeed, an even more recent study used a similar approach to investigate the role of PC firing irregularity in cerebellar functions of rhesus monkeys [50]. It has been suggested, both for mice and monkeys, that the irregularity of interspike intervals in PCs might play a role in the information transfer from the cerebellar cortex to DCN. Payne and colleagues used oculomotor behavior to test this hypothesis. Interestingly, while mean PCs firing varied accordingly to mean eye velocity, a strong correlation between these terms was not found observing moment-to-moment variations, which was therefore not related to spike irregularity. The optogenetic stimulation of PCs was able to independently control spike rate and irregularity, eliciting eye movements related to a linear rate code with 3–5ms temporal precision. These results are remarkable and show that the cerebellum exerts its control on movements using a firing code that works at impressively high rates, while a possible role of spike irregularity was not evident.

### 4.4. Movement Kinematics

The study of the role of specific neuronal types and pathways in the kinematics of movement execution is likely one of the fields that benefits more from the use of optogenetic tools. For example, it was recently shown that the transient suppression of spontaneous activity in a population of PCs was able to control movement kinematics in terms of modulation of size, speed, and timing [51]. The suppression of PC firing acts as a powerful mechanism that alters the precise control of movement, presumably due to disinhibition of target neurons in the DCN. Before the development of optogenetics, the precise and selective inactivation of PCs on behaviorally relevant timescales in vivo was impossible. In particular, in this study, the PC inhibition was achieved by activation of molecular layer interneurons (both stellate and basket cells), using a transgenic mouse line that expresses ChR2 in 85% of molecular layer interneurons, called nNos-BAC mice [52]. The transient suppression of PC activity in the eyeblink microzone resulted in the closure of the ipsilateral eyelid. In contrast, the same stimulation pattern in nearby locations of paravermal lobules IV-VI generated other orofacial movements such as mouth opening, cheek/lip lifting, and forward thrusting of the vibrissa. A change in the duration of the light pulse caused a corresponding change in the duration of eyelid closure, without affecting the blink speed. Both the intensity and the latency of the light pulse influenced the time needed to obtain the complete eyelid closure. These findings confirm previous observations that indicate the cerebellar cortex as playing a crucial role for adaptively controlling the timing of classically conditioned eyelid movements [19,53,54]. DCN neurons receiving inhibitory inputs from the PCs of the same module coherently showed excitatory responses that were modulated by the photostimulation intensity delivered to the cortex. Moreover, a strong linear correlation was found between the DCN neuron firing rate and both the size and speed of eyelid movements. This relationship suggests that PC firing and disinhibition-driven increases in DCN activity play a prominent role in the control of movement kinematics.

Another study used optogenetics to silence molecular layer interneurons by selectively expressing Archaerhodopsin 3.0 (Arch 3.0) using a Nos1-Cre transgenic mouse line and viral-mediated gene transfer (rAAV2-EF1a-DIO-eArch3.0-eYFP), while recording PC activity from the soma and the dendrites [55]. This experimental setting allowed the researchers to investigate PC activity in detail during self-paced locomotion. The optogenetic silencing of molecular layer interneurons was used to test whether the perturbation of excitation/inhibition balance in PCs affected self-paced locomotion. Indeed, optical stimulation induced consistent changes in locomotor behavior in the 70% of mice tested, leading to a slowdown or complete halt in locomotion. Changes in magnitude and duration observed were similar to those generated by direct optogenetic activation of PCs using Chr2 [56], suggesting the involvement of the same downstream motor-related pathways.

Together, these findings suggest that the regulation of PCs excitation/inhibition balance provides a sensitive modulatory mechanism of cerebellar output that is necessary for correct movement performance.

Another approach focused the attention on the cerebellar output, the DCN, and their modulation during locomotion [57]. Different studies have shown that movement, both as initiation and learning, is accompanied by an increase in DCN neuron firing, either by disinhibition or direct excitation [17,51,58,59,60,61,62,63,64]. As well as DCN silencing, inactivation or ablation has been related to prolonged reaction times or disruption of movements in behaving animals [65,66,67,68,69,70]. Sarnaik and colleagues performed extracellular recordings from PCs and DCN neurons (in the interpositus nucleus) while monitoring hind paw movement in awake, head-restrained mice running on a freely rotating treadmill (Figure 2). Optogenetics were used to modulate PC activity in mice offspring of *Ai27D* x *Pcp2*-cre crosses expressing ChR2 in PCs with the use of the viral constructs AAV9.CBA.Flex.ChR2(H134R)-mCherry.WPRE.SV40 or AAV9.EF1a.DIO.hChR2(H134R)-eYFP.WPRE.hGH [57]. Photostimulation was used to modify the rate and temporal pattern of the inhibition of DCN neurons, at different times of the stride cycle during locomotion on the treadmill. The fast timing kinetics of perturbations pemitted by optogenetics was used to find clear correlations between DCN activity perturbation and alterations of the observed movements (as “slips”). The direct stimulation of PCs in awake mice is then sufficient to provoke slips during voluntary locomotion, in a way that is related to DCN inhibition mediated by PCs. A correlation was found between locomotion transient disruptions and the modulation of DCN activity, rather than the absolute firing rate. These results suggest that the role of DCN in locomotion goes beyond the transmission of incoming inputs to downstream regions, providing further details on how the cerebellar output actively regulates locomotion. While DCN inactivation consistently disrupted ongoing movements, dampening the DCN activity still had the effect of causing irregularities of movements during locomotion.

### 4.5. Movement Disorders

The cerebellum is notoriously involved in a number of motor disorders, involving alterations of cerebellar processing or of cerebellar connections to other brain regions. Optogenetics is evidently helpful in these cases in order to understand the specificity of the alterations at the level of neuronal types or pathways involved. For example, a recent study described the cerebellar role in modulating the activity of basal ganglia neurons in the striatum [71]. Basal ganglia and cerebellum are brain structures fundamental for the initiation and correct execution of voluntary movement [72,73], and impairment in their function is responsible for several motor disorders, such as Parkinson’s disease and ataxia [74]. The optogenetic activation of the cerebellum (using AAV2-Syn-ChR2(H134R)-YFP injected in the dentate nucleus) proved capable of altering striatal neuron activity in half the cases of freely moving mice [71]. This effect had a short latency (around 10ms), though it involved an indirect pathway through the thalamus and affected cortical-driven plasticity in this region. Beyond this, Chen and colleagues characterized the cerebellar influence on basal ganglia activity in a mouse model of cerebellar-induced dystonia (obtained with ouabain infusion, to mimic rapid onset dystonia Parkinsonism). In this model, cerebellar-induced dystonic postures were correlated with abnormal neuronal activity in the dorsolateral striatum. Interestingly, the alteration in cerebellar output was transmitted through the thalamic station (intralaminar nuclei) and altered basal ganglia activity. Silencing this thalamic nucleus using optogenetics (AAV2/5-*CAG*-ArchT-GFP) alleviated the dystonic symptoms.

The cerebellar role in motor diseases involving brain circuits were recently investigated for Parkinson’s disease, revealing a reorganization of cerebellar–cerebral circuit in a mouse model of this disease, causing a lesion of midbrain dopaminergic neurons [75]. Interestingly, though bearing alterations at the cerebellar level, this mouse model did not show any motor symptoms usually associated to cerebellar dysfunction, suggesting normal coupling to descending pathways. Indeed, the alterations in the cerebellum were described at the level of populations of PCs showing slow and irregular firing, compared to controls, likely explaining the increased activity of DCN neurons [75]. Cerebellar stimulation elicited weaker electrocortical responses in the motor cortex of lesioned animals, in agreement with the hypothesis of a decrease in cerebellar modulation of motor cortex activity in Parkinson’s disease [76,77,78]. These observations suggest a putative role of the cerebellum in Parkinson’s disease tremor [79,80].

Though the details of cerebellar coordination of multi-joint movements are still debated, new clues were derived from an investigation of these mechanisms in the ataxic and dystonic mouse (the tottering mouse). The olivo-cerebellar system is thought to play a major role, through climbing fibers generating complex spikes in PCs. The inferior olive displays oscillatory dynamics that have been proposed to take part in controlling the motor output [81,82]. The tottering mouse is characterized by reduced levels of complex spikes synchronicity, delayed compared to the initiation of movement, likely affecting the motor deficits. A recent study exploited this model and optogenetics to investigate the role of synchronicity of PC activation in microzones [56]. This study exploits the fast kinetics of photostimulation and the possibility to select its spatial extension. The authors reported that synchronous activation of PC ensembles could facilitate both the initiation and coordination of movements, showing that simple spikes in PCs play a major role. The mechanisms highlighted in this study are likely the same as those involved in multi-joint movement coordination.

On a related note, the cerebellum was reported to play a role in post-stroke rehabilitation. This effect is likely to be mediated by the connections from the cerebellum to various cerebral cortex areas. A recent paper addressed this issue in a mouse model with induced photothrombotic stroke in the sensorimotor cortex [83]. Indeed, it was reported that the stimulation of the dentate nucleus in rodents positively affected post-stroke motor recovery [84]. Zhang and colleagues demonstrated that the effectiveness of environmental enrichment, known to promote rehabilitation [85], is facilitated when paired with cerebellar stimulation. To obtain the stimulation specificity, they photo-activated the DCN, previously injected with rAAV-hsyn-hChR2-mcherry-WPRE-PA. Notably, this investigation also demonstrated that the cerebellum is necessary to mediate the positive effects of environmental enrichment on rehabilitation, by inactivating the DCN using optogenetics (rAAV-hsyn-eNPHR3.0-mcherry-WPRE-PA). It is likely that the cerebellar role in post-stroke rehabilitation relies on cerebellar contribution to cortical plasticity, an issue still debated, but sustained by accumulating evidence [86,87,88,89]. This paper is a good example of the advancements made possible by optogenetics: obtaining stimulation precision, exploring the possibility to evaluate the effects of activation and inhibition of the same neurons, and achieving temporal accuracy. As the authors themselves stated, future investigations will address the role of single cerebellar neuronal types in these mechanisms.

## 5. Non-Sensorimotor Functions 

While the cerebellar role in sensorimotor functions and motor coordination is well established, the nature of its impact on cognition and emotion remains more difficult to address. The connections involved are usually indirect and the convergence of more inputs to associative areas makes it incredibly difficult to detect the specific role of the cerebellum among the contributions of other brain areas. To further complicate the picture, the behavioral counterpart to the cerebellar involvement in non-sensorimotor functions is difficult to retrieve. The understanding of the cerebellar involvement in non-sensorimotor functions was first prompted by the development of functional magnetic resonance imaging (fMRI), during the past 25 years. Recent anatomical, structural and functional evidence has revealed that cerebellar activation is associated with addiction, social cognition and emotional processing [90,91,92,93]. Furthermore, cerebellar lesions are implicated in cognitive disorders and abnormal social behavior such as in autism spectrum disorders (ASD), cognitive affective syndrome, schizophrenia and epilepsy [93,94,95,96,97,98,99,100]. Notably, ASD patients report both motor dysfunctions together with non-motor symptoms, suggesting that cerebellar impairment might indeed contribute to both [95,100,101], likely depending on the connected brain areas [102]. In non-human primates, tract-tracing investigations have demonstrated that topographically distinct regions (called output channels) of the DCN project to different cortical areas: with dorsal parts sending efferent fibers to the motor cortex and ventral parts to the prefrontal and parietal cortexes, which are generally involved in cognitive and higher-order executive functions [103,104]. Recently, neuroimaging studies have reported a similar connectivity topography in human DCN [105,106]. The prefrontal cortex and its extensive connections with other cortical, subcortical and brain stem areas have extensively been investigated. These studies provided evidence for two different pathways through which DCN communicate with the prefrontal cortex. The main pathway involves glutamatergic neurons located in the DCN, which connect to primary thalamic nuclei such as ventrolateral, ventromedial and, additionally, centrolateral nuclei [103,107,108]. The connectivity among the DCN and thalamic nuclei is yet to be fully characterized. It is unclear whether the cerebellum projects to the mediodorsal thalamic nuclei, which are known to provide the main thalamic inputs to the prefrontal cortex [109,110]. Recently, reciprocal connectivity between the prefrontal cortex and ventral thalamic nuclei (specifically ventromedial) has been shown [109,111,112]. The second pathway involves DCN projections to the ventral tegmental area (VTA) which, in turn, send dopaminergic fibers to the prefrontal cortex [113]. A growing body of evidence has reported that VTA dopaminergic projections in the prefrontal cortex, in addition to influencing stress-related function and working memory [114,115], mediate many of the higher-order cognitive functions, including reward, motivation, attention and behavioral flexibility [116,117,118,119]. Notably, alterations in dopaminergic neurotransmission in the prefrontal cortex has been shown in a number of patients diagnosed with schizophrenia and autism [120,121,122]. Recently, it has been shown that the electrical stimulation of DCN was able to indirectly evoke the release of dopamine in the prefrontal cortex (dentate-reticulotegmental-peduncolopontine-VTA-prefrontal cortex pathway) [123,124], suggesting a cerebellar contribution to reward driven behaviors.

The following sections summarize the recent findings in the main non-sensorimotor functions addressed using optogenetics (see Table 1).

### 5.1. Reward and Social Behavior

A recent report exploited optogenetic specificity to demonstrate the involvement of the cerebellum in non-motor functions through a direct pathway. This is particularly interesting, as cerebellar impact on non-sensorimotor functions is usually considered indirect, passing mainly through the thalamic relay. In this work, Carta and colleagues reported the presence, the physiological nature and the functional significance of the direct connectivity between DCN and the VTA in mice [125]. In particular, the adeno-associated virus carrying the ChR2 (AAV1-hSyn-ChR2-YFP) or the archaerhodopsin (AAV5-*CAG*-ArchT-GFP) was injected into the DCN (Figure 3). The optogenetic modulation of DCN axons activity was achieved by illuminating the infected fibers directly in the VTA in acute slices. This allowed them to rule out the spurious activation of other fibers and it provided a unique tool to determine the monosynaptic origin of the responses detected in the VTA. This same investigation showed that DCN axons form glutamatergic synapses with both dopaminergic and non-dopaminergic neurons in the VTA and that their optogenetic activation directly increased the activity of VTA neurons, both ex vivo and in vivo. While the experiments in slices allowed to characterize the high efficacy of the transmission at these connections, behavioral tests were key to determine the functional significance of this pathway and the effects of cerebellar activity on VTA-dependent behavior involving reward and sociability. Interestingly, the optogenetic activation of DCN axons in the VTA was sufficient to cause long-term place preference, while their inhibition completely blocked social preference in the three-chamber sociability test [125]. Though this test is known to also involve the prefrontal cortex [132], the use of optogenetics to activate or inhibit DCN axons in the VTA allowed the researchers to prove a causal relationship between cerebellar activity and VTA-driven motivated behavior. This experimental setting provided the demonstration of a direct, functional, glutamatergic pathway from the DCN to the VTA, actively contributing to motivated behavior, further expanding the view of cerebellar role in non-motor functions [93].

### 5.2. Working Memory and Decision-Making

Working memory is a fundamental part of the decision-making process. These are fundamental cognitive functions involving distributed interacting network of different brain areas, with the prefrontal cortex at the core [133]. The cerebellar contribution to working memory has been reported by several studies in humans. Cerebellar damages or dysfunctions often lead to working memory impairments [3,134,135,136]. In order to investigate this aspect of cerebellar function in behaving mice, Deverett and colleagues exploited optogenetics to test whether direct, precise and transient disruption of cerebellar activity modulated the accumulation of sensory information in working memory [129]. Mice expressing ChR2 in PCs were obtained by crossing mice with genotypes *Pcp2*-Cre for PC specificity and *Ai27D* for ChR2. The authors showed the optogenetic manipulation of PC-impaired decision making by reducing the ability to effectively retain past information in working memory (Figure 4). In particular, they tested the accumulation of sensory information using a behavioral task for head-fixed mice. In each trial, the animal received air-puffs directed both on the left and right whisker pads, simultaneously, for seconds (sensory evidence presentation). After a delay, an auditive cue signaled that the mouse could retrieve a water reward, either by licking a left- or right-placed port. The task was successful, and the reward delivered, when the licking direction was the same as the whisker pad which received more air-puffs (see Figure 4 for a schematic representation of this task). ChR2-expressing PCs were stimulated during the sensory evidence presentation, preceding the decision, and led to reductions in performance accuracy. In this case, the use of optogenetics allowed the researchers to narrow down the specificity of the effect on behavior to a single neuronal type, the PCs, that provide the sole output of the cerebellar cortex. While the use of mutant mice could have helped identifying an important role of these neurons, optogenetics provided the advantage to modify neuronal activity in a transient way. This allowed them to modify neuronal activity in only the temporal window significant for the accumulation of the sensory evidence in working memory. The transient and reversible nature of the optogenetic interference with neuronal activity allowed to rule out possible motor impairments leading to failing the test and reinforcing the concept that the cerebellar cortex is indeed involved in sensory evidence retainment for working memory. These results may help account for the many clinical findings linking cerebellar activity to working memory and decision making in humans [103,136,137,138]. Since working memory is considered dependent on the activity of forebrain regions [139,140], it is likely that the cerebellar role described above is mediated by indirect cerebellar–neocortical projections. Further investigations on the subject would definitely benefit from optogenetics use to dissect cerebellar pathways.

### 5.3. Schizophrenia and Cognition

Cerebellar projections to the frontal cortexes, passing through the thalamus and the VTA, are considered the base of its role in cognitive performances and related diseases, such as autism and schizophrenia [102]. The cerebellum, thalamus and frontal cortex have been found to be involved in schizophrenia, leading to a theory of a distributed network impairment at the core of this disease. Interestingly, the electrical stimulation of the cerebellar vermis using MRI-guided transcranial magnetic stimulation (TMS) proved to be a successful treatment in patients with refractory schizophrenia [141,142,143]. This evidence is of particular relevance since only few treatment options are available for this neurological disorder. Parker and colleagues proposed a novel hypothesis that cerebellar stimulation could improve cognitive symptoms of schizophrenia by normalizing prefrontal cortex activity, suggesting an experimental design combining optogenetics, pharmacology, and electrophysiology tools in awake-behaving animals to investigate the cerebellar role in cognitive processes [144,145]. The injection of optogenetic constructs in the DCN allowed to specifically activate the cerebellar inputs to the VTA and different thalamic nuclei. The schizophrenia condition is mimicked by pharmacologically disrupting D1-mediated dopamine signaling in the prefrontal cortex in rodents, to reproduce the effect of reduced expression of D1-like dopamine receptors observed in the same brain area in human patients, considered at the core of schizophrenia dysfunctions as deficits in internal timing behavior [131,146,147,148]. In these animals, optogenetic stimulation of ChR2-expressing DCN axons in the thalamic ventrolateral nuclei rescued the performance in an interval timing cognitive task, found impaired by disrupting D1 dopamine signaling [131]. Moreover, the prefrontal cortex neurons showed an increase in the firing rate following optogenetic stimulation of DCN projections in the thalamus. These data support the idea that the cerebellar circuit can compensate for dysfunctional prefrontal cortex activity in schizophrenia-mimicking conditions [131]. The use of optogenetics in this experimental settings allowed for the matching of the two formerly separated observations that D1-like dopamine receptors in the prefrontal cortex are involved in timing processes [145] and the general knowledge that the cerebellum is a timing-machine operating in the sub-millisecond time scale [149,150,151], likely contributing to the internal timing task.

### 5.4. Temporal Lobe Epilepsy and Absence Seizure

Temporal lobe epilepsy (TLE) is the most diffused focal epilepsy in adults. TLE patients are often unresponsive to anti-epileptic drugs and not suitable for surgical intervention. In this type of epilepsy, seizures often start at the hippocampal level. Nevertheless, a direct intervention on hippocampal neurons activity proved inefficient to block the seizures. Over the last few decades, electrophysiological and neuroimaging studies have provided evidence for a cerebellar role in influencing cerebral epileptic activity. For example, seizures can arise directly from cerebellar structures [152,153] and changes in cerebellar blood flow, and electroencephalogram and neuronal activity have been reported during seizures [154,155,156,157]. Despite this evidence, experimental investigations using animal models of TLE to unravel the role of the cerebellum in the epileptic phenomenon have produced mixed results. The lack of specificity inherent with electrical stimulation was most likely responsible for these unclear conclusions. A recent study using mice lacking cerebellar long-term depression showed that information processing in hippocampal place cells is disrupted [158], and suggested the cerebellum as a new target for intervention in TLE. Thus, Krook-Magnuson and colleagues tested whether optogenetic intervention targeting the cerebellum could provide seizure control in TLE, exploiting the specificity achievable with this technique [126]. In particular, the unilateral intrahippocampal kainate mouse model of TLE was used. ChR2 and NpHr (light-driven chloride pump halorhodopsin) were used in different experiments to test the effects of activation or inhibition of parvalbumin-expressing neurons, including PCs, in the cerebellar cortex. In this case, the specificity of optogenetics was not confined to a single neuronal type, since also molecular layer interneurons might express parvalbumin. Nevertheless, the authors could draw noticeable conclusions on the effect of activating or silencing cerebellar cortical neurons and these results prompted further research. In particular, they achieved the puzzling result that the seizure duration in the hippocampus was somehow reduced by both the activation and inhibition of parvalbumin-expressing neurons in the cerebellar cortex, suggesting that the direction of modulation was not critical. The authors comment that these apparent contradictory results can arise from the fact that parvalbumin is also expressed in molecular layer interneurons, and that the net effect on DCN is therefore difficult to infer. This is in agreement with a recent report using a pan-neuronal promoter for ChR2 expression in the cerebellar molecular layer while recording from the DCN, showing the mixed effect of light activation likely depending on the prevalent effect of direct activation of PCs or inhibition through molecular layer interneurons [159]. Krook-Magnuson and colleagues further exploited optogenetic specificity by replicating their key findings in a mouse line expressing ChR2 selectively in PCs. Their results showed that selective excitation of PCs was not only capable of reducing hippocampal seizure duration but also of inhibiting seizure induction (increasing the time between seizures) [126]. In conclusion, the use of optogenetic approaches was key to demonstrate that the cerebellum can be an effective target to inhibit hippocampal seizures. Moreover, the complex response observed to optogenetic manipulation of PCs offers one possible explanation for the fact that the direction of modulation (excitation/inhibition) was not critical for hippocampal seizure inhibition. During optogenetic excitation of PCs, DCN show decreased firing rates [63,160,161]. However, PC excitation is subsequently followed by a pause in PC activity [126], in agreement with the burst–pause behavior typical of these neurons [162], and a corresponding increase in firing in the DCN [161]. Therefore, both optogenetic inhibition and excitation of PCs could result in excitation of the DCN and thereby the cessation of hippocampal seizures. To verify this hypothesis, Streng and Krook-Magnuson optogenetically stimulated glutamatergic neurons in the DCN (mainly in the fastigial nucleus) during hippocampal seizures in a mouse model of TLE [127]. Indeed, optogenetic inhibition of fastigial nuclei neurons had no effect on hippocampal seizures while excitation robustly attenuated them. These results strongly suggest that DCN excitation was responsible for the decrease in hippocampal seizures observed while modifying the activity in the cerebellar cortex. Addressing DCN activity might therefore be key to successful intervention on TLE.

The role of the cerebellum in epileptic phenomena was also proved by recent work on cerebellar contributions to absence epilepsy [128]. Absence epilepsy is characterized by sudden periods of impaired consciousness which typically last up to ten seconds, associated to behavioral arrest. The electrophysiological counterpart was found in thalamocortical oscillations due to excessive activation of the cerebral cortex, originating the so-called generalized spike-and-wave discharges (GSWDs). Kros and colleagues tested whether altering the output of the cerebellum might affect thalamocortical oscillations in the tottering mouse, characterized by a missense mutation of the *Cacna1a* gene leading to a loss-of-function of calcium channels, and considered an established model for absence epilepsy. To modify DCN activity, the virus containing the ChR2 (AAV2-hSyn-ChR2(H134R)-EYFP) was injected into the DCN. The specificity of the optogenetic approach was, in this case, obtained using localized injections. This experimental setting allowed the researchers to determine the effect of cerebellar activation on GSWDs, recorded through electrodes implanted in the primary motor and sensory cortexes. The results showed that optical activation of DCN neurons significantly reduced or even stopped GSWDs within 150ms from the onset of stimulation (both bilateral and unilateral). The authors showed that the optogenetic activation of DCN neurons was able to alter the spontaneous activity of these neurons, often involved in oscillatory cycles. Interestingly, the precise temporal efficiency of optogenetic stimulation was ideal to explore the temporal window of the GSWD most responsive to DCN activation, showing that the most effective results were obtained when stimulation was applied during the “wave” phase of GSWD, that is, when cortical neurons are normally silent. In this paper, optogenetic stimulation was irreplaceable for both the specificity of DCN activation and for the temporal precision involved in these kinds of investigations.

### 5.5. Control of Blood Pressure

The cerebellar lobule IX of the vestibulocerebellum (also called the uvula) is known to be extensively connected to brainstem regions involved in vestibular signal processing and, interestingly, cardiovascular regulation. Thus, the uvula could take part in the reflex mechanisms that intervene during postural alterations to homeostatically regulate blood pressure. Indeed, previous investigations have shown that lesions of this cerebellar region are able to impair the baroreceptor reflex and cardiovascular responses during postural alterations [163,164]. PCs in the uvula project, indirectly, to the solitary tract nucleus, which receives primary afferent fibers from the baroreceptors. Though this pathway was the best candidate to explain the experimental evidence, no clear causal connection could be detected between PC activity and cardiovascular responses to postural alterations. Tsubota and colleagues [130] addressed this issue by infecting rat cerebellar uvula with a lentivirus carrying the eNpHR, selectively expressed in PCs due to the specificity of the *L7* promoter (Lenti-*sL7*-eNpHR-EYFP-WPRE). By inhibiting PCs at different times during the postural alterations (consisting in 30° head-up or head-down tilts), this study clearly showed that PC activity in the uvula is involved in the regulation of blood pressure in response to postural alterations (in particular during head-down tilts) and also in recovery to baseline levels. Previous studies showing the uvula role in these mechanisms were performed by lesioning the cerebellar areas involved [164]. The optogenetic approach provides several advantages, as the cell-type specificity, the transient nature of neuronal silencing, the temporal modulation of the inhibition (that lasted from 1s to several seconds in Tsubota’s study) and the possibility to avoid neural compensations as consequences to the lesion. The investigation summarized in this section is conducted on anesthetized rats, unlike all the other reports taken into consideration in this review. Nevertheless, the study by Tsubota and colleagues was considered here due to the notable advances provided by the use of optogenetics and to the peculiarity of the topic addressed. Beyond its role in sensorimotor and non-sensorimotor functions, the cerebellum is involved in homeostatic reflexes as the baroreceptor one, controlling blood pressure through PC activity. Optogenetics is helping us to unravel these less investigated mechanisms.

## 6. Clinical Aspects 

Cerebellar stimulation as a clinical tool to reduce patients’ symptoms of motor and non-motor diseases is gaining more and more attention as of late. For a comprehensive review of this topic, see [165]. In the context of this review, it is interesting to point out that optogenetic stimulation of the cerebellum could provide a suitable tool to overcome some of the issues raised by the use of Deep Brain Stimulation, Transcranial Direct Current Stimulation, and Transcranial Magnetic Stimulation. First of all, optogenetics can guarantee specificity of the target zone. Where specific markers are available, even single neuronal types might be selectively targeted. This would avoid the unspecific and unforeseeable effects of stimulating the cerebellar cortex, for example. Secondly, optogenetics provide the possibility to either activate or inhibit neuronal activity. At the moment, inhibition can only be achieved through lesioning or surgical ablation of brain tissue. Thirdly, optogenetics is ideal to provide neuronal activation/inhibition in precise and sharp temporal windows, that can be modulated down to the milliseconds time scale. Interestingly, this aspect of optogenetics is particularly suitable to be paired to systems involving brain–computer interface (BCI) technology. An example is reported in one of the papers cited in the previous section on absence epilepsy [128]. Kros and colleagues found that optogenetic intervention on the DCN in the tottering mouse was more efficient in the “wave” phase of the GSWD. The authors suggested that the use of a BCI in this case might be useful in order to provide stimulation to the DCN in the most effective way, and they indeed provided a proof of principle of the validity of this procedure in their experimental conditions. The future of brain stimulation will most likely involve the cerebellum on various motor and non-motor pathologies, and will be radically improved wherever the implementation of optogenetics will be successful. Indeed, recent constructs show reduced vector-associated cytotoxicity, suggesting that a clinical application in substitution to deep brain stimulation might be feasible soon. At the moment, optogenetics is applied only to vision restoration [166], but its overcoming of previous technical issues [167] might open several clinical applications in the field of cerebellum-related pathologies, also.

## Figures and Tables

**Figure 1 ijms-21-02494-f001:**
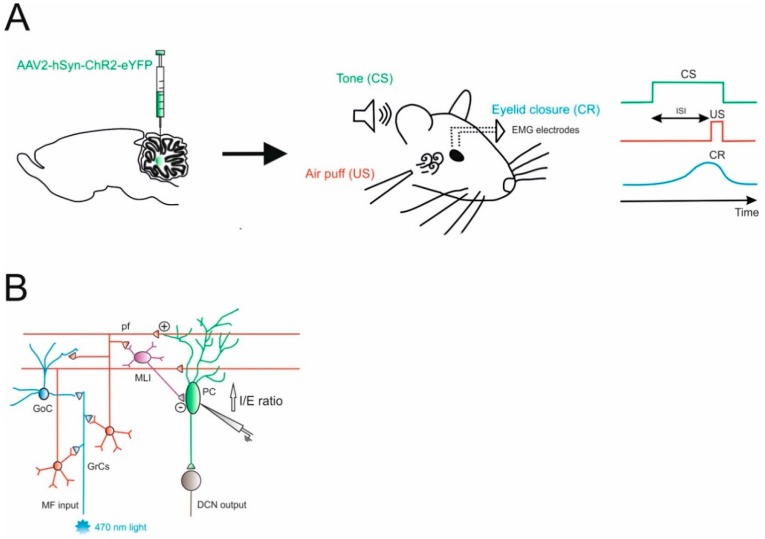
Role of Excitatory Cerebellar Nucleocortical Circuit in controlling associative motor learning. (**A**) *Left*, schematic viral injection of AAV2-hSyn-ChR2-eYFP targeting the interposed nucleus. *Right*, in the eyeblink conditioning (EBC) paradigm, a conditioned stimulus (CS, tone) is presented before the onset of an unconditioned stimulus (US, air puff). The inter-stimulus interval (ISI) is the time interval between CS and US onset. The two stimuli normally co-terminate. During the training, mice learn to generate a delayed conditioned response (CR) to the CS before the onset of the expected US. (**B**) Mossy fiber (MF) afferents can influence deep cerebellar nuclei (DCN) output via projections onto granule cells (GrCs). GrCs can directly affect Purkinje cell (PC) activity via parallel fiber connections or indirectly via feedforward GrC-MLI-PC processing. The EBC paradigm increases the inhibitory/excitatory (I/E) ratio in PCs, indicating a preferential enhancement of the feedforward inhibitory GrC-MLI-PC pathway.

**Figure 2 ijms-21-02494-f002:**
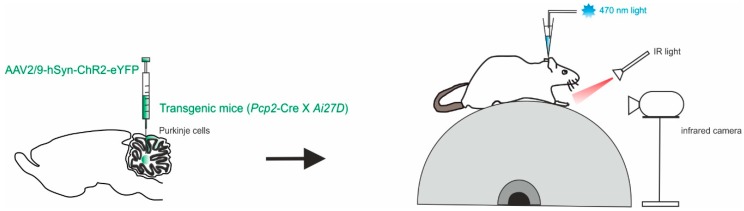
Modulation of PCs and DCN during voluntary locomotion. *Left*, in *Pcp2*-cre mice, a mixture of viruses (AAV2/9) was injected to express ChR2 with a fluorescent marker conjugated with eYFP in the interpositus nucleus. *Right*, schematics of the setup with a head-fixed mouse running on the cylindrical treadmill. Paw movements are recorded with an infrared camera. The patch pipette contained an electrode wire and an optical fiber.

**Figure 3 ijms-21-02494-f003:**
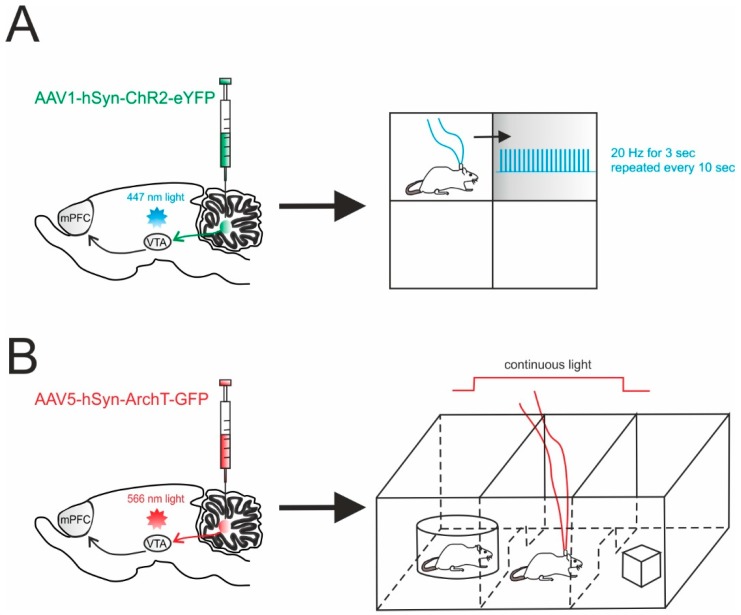
Cerebellar influence on reward and sociability. (**A**) *Left*, schematic viral injection of AAV1-hSyn-ChR2-eYFP targeting the deep cerebellar nuclei (DCN). Optical fibers were implanted into the ventral tegmental area (VTA), bilaterally. *Right*, mice were allowed to explore a square chamber, divided in quadrants, one of which was associated to a reward. Whenever the mouse entered in the reward quadrant, cerebellar axons innervating the VTA were optically stimulated. For the time the mouse spent in the reward quadrant, the stimulus (20 Hz for 3 s) was repeated every 10 s. (**B**) *Left*, AAV5-hSyn-ArchT-GFP was injected in the DCN in order to selectively inhibit cerebellar axons through bilateral optical fibers implantation into the VTA. *Right*, Social preference was examined using a three-chambered social task. The tested mouse was allowed to approach a “stimulus” mouse confined to one side chamber (social side) or a novel object placed on the other side (non-social side). On the first day (training day), the mouse was allowed to freely explore all three chambers. On the subsequent day, the cerebellar axons innervating the VTA were optically inhibited for the time the mouse stayed by the social side. The optogenetic stimulation was immediately ceased when the mouse exited the social side.

**Figure 4 ijms-21-02494-f004:**
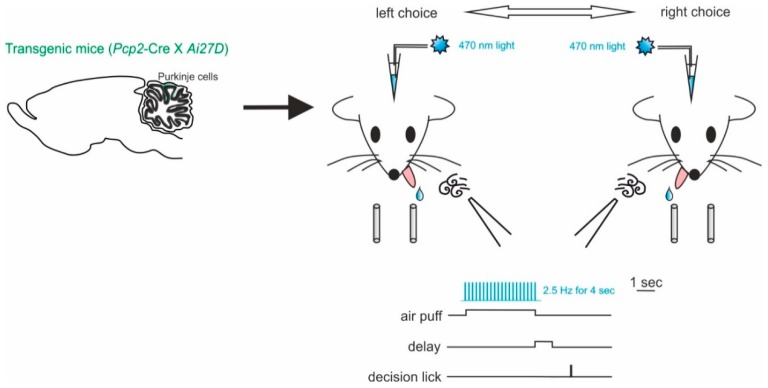
Impaired decision-making by disruption of cerebellar activity during sensory evidence accumulation. *Left*, mice of genotypes *Pcp2*-Cre for PC specificity and *Ai27D* for ChR2 were used. ChR2-expressing PCs were stimulated using light delivered through optical fibers implanted bilaterally over crus I of the cerebellum. *Right*, schematic of the evidence-accumulation decision-making task. In each trial, two streams of randomly timed air puffs were delivered to the left and right whiskers. After an 800-ms delay, mice licked one of two lick ports indicating the side with more cumulative puffs to receive a water reward.

**Table 1 ijms-21-02494-t001:** Viral constructs used in optogenetic manipulation of non-motor behavior.

Serotype	Promoter	Expression (Cell type)	Opsin	Cerebellar Region	Behavior	Behavioral Outcomes
AAV1	hSyn	Neuron-specific	ChR2	DCN	Reward	Increased place preference [125]
AAV5	*CAG*	All cells	ArchT	DCN	Social behavior	Altered social preference [125]
	*Pcp2*-Cre *	Purkinje cells	ChR2	Cortex (simplex)	Epilepsy	Reduction in hippocampal seizure duration and seizure-induced inhibition [126]
AAV9	*VGluT*-Cre *	Glutamatergic	ChR2	DCN (fastigial)	Epilepsy	Reduction in hippocampal seizure duration [127]
AAV2	hSyn	Neuron-specific	ChR2	DCN (dentate)	Epilepsy	Reduction in thalamocortical oscillations [128]
AAV2/9	*Pcp2*-Cre *	Purkinje cells	ChR2	Cortex (Crus I)	Working memory	Reduction in performance accuracy [129]
Lentivirus	*L7*	Purkinje cells	eNpHR3.0	Cortex (uvula)	Postural alterations	Reduction in the extent of blood pressure recovery [130]
AAV	*CamKII*	Glutamatergic	ChR2	DCN (dentate)	Schizophrenia	Increase in prefrontal activity [131]

Transgenic lines that have been used in mouse studies are denoted by an asterisk (*).

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
