# Peer review of "The Optogenetic Revolution in Cerebellar Investigations"

_ijms, 2020, doi:10.3390/ijms21072494_

Round 1
Reviewer 1 Report
This is a very timely and interesting review presented by the D’Angelo and Mapelli labs. There is a sharp increase in the number of optogenetic / photostimulation studies on cerebellar function, and it is almost impossible to keep track of these fast developments. This review nicely summarizes what has been done so far, and points toward interesting new avenues of research.
Major comment:
- 2: when discussing pros and cons of optogenetic techniques, the authors seem to omit the possibly most important limitation. Optogenetics is an excellent tool from a qualitative perspective (specific regions, cell types), but poorly replaces physiological activity patterns quantitatively. For example, when channelrhodopsin is photo-activated in a defined structure such as the DCN, it will activate the entire DCN, or ‘blocks’ of it, while under physiological conditions very specific and limited subsets of neurons might be activated that may be intermingled with non-active cells. Similarly, it is very hard to mimic the proper kinetics of activation. This quantitative limitation should be discussed in detail.
Minor comments:
- Abstract, l.12: ‘….brought investigation to an entirely new level’. This statement appears somewhat exaggerated and unprecise. It might be better to emphasize that optogenetics added new capabilities.
- 4, l. 151: for intrinsic plasticity functions in EBC, also cite Grasselli et al, PLoS Biology 18 (2020).
- 9, l. 381: the separation between motor and non-motor functions does not apply as clearly for autism as the authors state. In fact, there are interesting studies on motor learning deficits in autism mouse models that the authors should refer to here. This is actually a nice point to make, because the very same cerebellar processing deficits (from a cellular point of view) might contribute to motor and cognitive consequences of cerebellar dysfunction in autism, depending on the target structure.
- Table 1: the authors should include the references for the studies listed.
Author Response
We thank the Reviewer for the positive comments. We addressed every major and minor comments proposed, which considerably helped in improving the manuscript.
Major comment:
2: when discussing pros and cons of optogenetic techniques, the authors seem to omit the possibly most important limitation. Optogenetics is an excellent tool from a qualitative perspective (specific regions, cell types), but poorly replaces physiological activity patterns quantitatively. For example, when channelrhodopsin is photo-activated in a defined structure such as the DCN, it will activate the entire DCN, or ‘blocks’ of it, while under physiological conditions very specific and limited subsets of neurons might be activated that may be intermingled with non-active cells. Similarly, it is very hard to mimic the proper kinetics of activation. This quantitative limitation should be discussed in detail.
We thank the Reviewer for this comment. A clear statement on this limitation was not present in the original manuscript. The following paragraph has been added to the text (from line 124).
“Before proceeding, another critical issue is worth pointing out. Optogenetics is indeed a revolutionary method to excite or inhibit neurons, since it involves the opening of ion channels on their membranes, generating ion fluxes to modify membrane voltage. This condition is very similar to the physiological processes involved during neuronal activity, but it is essential to keep in mind that optogenetic stimulation is very different from the physiological condition. It is not possible to have control over the amount of currents induced in a single neuron or to affect only those neurons that physiologically are activated together by a common pathway. Every neuron expressing the opsins will react when illuminated, therefore activating or inhibiting entire regions. This condition is quantitatively different from physiological activation in terms of currents amplitude in single neurons and the number of neurons affected. Though this specification was necessary, a stimulation method acting directly on neurons and mimicking the exact physiological activation is not available at the moment”.
Minor comments:
Abstract, l.12: ‘….brought investigation to an entirely new level’. This statement appears somewhat exaggerated and unprecise. It might be better to emphasize that optogenetics added new capabilities.
That statement was indeed too emphatic and was re-written according to the Reviewer’s suggestion.
4, l. 151: for intrinsic plasticity functions in EBC, also cite Grasselli et al, PLoS Biology 18 (2020).
We thank the Reviewer for bringing to our attention this paper, which is now cited as suggested (l. 189).
9, l. 381: the separation between motor and non-motor functions does not apply as clearly for autism as the authors state. In fact, there are interesting studies on motor learning deficits in autism mouse models that the authors should refer to here. This is actually a nice point to make, because the very same cerebellar processing deficits (from a cellular point of view) might contribute to motor and cognitive consequences of cerebellar dysfunction in autism, depending on the target structure.
We never intended to state that autism is a pure non-motor disease. It is enlisted in the non-sensorimotor section because of the cerebellar role in non-motor symptoms typical of this condition, investigated using optogenetics. We agree with the Reviewer that this was not evident in the original text. A sentence and new references have been added to state this point (from line 421).
Table 1: the authors should include the references for the studies listed.
The references are now cited in the Table.
Reviewer 2 Report
Review
The optogenetic revolution in cerebellar investigations (Manuscript ID: ijms-761057).
Having carefully read this manuscript, my view is that it is interesting and is well written. The topic is timely. This review has also a basic importance in the clinical and experimental field of neurodegenerative diseases. This paper should be published in “International Journal of Molecular Sciences”.
However, in spite of not having detected any major problems, I would nevertheless suggest that the authors bear in mind two items:
1.- A list of abbreviations should be included. Is it possible?
2.- Previous to the section number 2. This is entitled "2. Pros and Cons of optogenetics", a section tentatively entitled "overview of the cerebellum" might be included. In this section the authors can describe the cerebellum, i.e., vermis and hemispheres, cerebellum cell types,........As example:
1.- Marzban, H., Del Bigio, M. R., Alizadeh, J., Ghavami, S., Zachariah, R. M., Rastegar, M. (2015) Cellular commitment in the developing cerebellum. Front Cell Neurosci 12, 8: 450. DOI: 10.3389/fncel.2014.00450.
2.- Leto, K., Arancillo, M., Becker, E.B., Buffo, A., Chiang, C., Ding, B., … Hawkes, R. (2016). Consensus paper: cerebellar development. Cerebellum, 15, 789-828. https://doi.org/10.1007/s12311-015-0724-2.
Author Response
We thank the Reviewer for the positive comments on our manuscript and for the helpful suggestions that we believe significantly improved the final version of the text.
1.- A list of abbreviations should be included. Is it possible?
A list of abbreviations has been added to the text. Indeed, organizing this list allowed us to improve the use of abbreviations and acronyms throughout the manuscript.
2.- Previous to the section number 2. This is entitled "2. Pros and Cons of optogenetics", a section tentatively entitled "overview of the cerebellum" might be included. In this section the authors can describe the cerebellum, i.e., vermis and hemispheres, cerebellum cell types,........As example:
1.- Marzban, H., Del Bigio, M. R., Alizadeh, J., Ghavami, S., Zachariah, R. M., Rastegar, M. (2015) Cellular commitment in the developing cerebellum. Front Cell Neurosci 12, 8: 450. DOI: 10.3389/fncel.2014.00450.
2.- Leto, K., Arancillo, M., Becker, E.B., Buffo, A., Chiang, C., Ding, B., … Hawkes, R. (2016). Consensus paper: cerebellar development. Cerebellum, 15, 789-828.
We thank the Reviewer for this useful suggestion.
An overview of cerebellar anatomy and microcircuits organization is now present as section 2. We believe that this will improve manuscript clarity.